# The Role of Maximal TURBT in Muscle-Invasive Bladder Cancer: Balancing Benefits in Bladder Preservation and Beyond

**DOI:** 10.3390/cancers16193361

**Published:** 2024-09-30

**Authors:** Farshad Sheybaee Moghaddam, Sami Dwabe, Nataliya Mar, Leila Safdari, Navin Sabharwal, Hanan Goldberg, Michael Daneshvar, Arash Rezazadeh Kalebasty

**Affiliations:** 1Institute of Urology, University of Southern California, Los Angeles, CA 90089, USA; farshad.moghaddam@med.usc.edu; 2Department of Medicine, University of California Irvine, Orange, CA 92868, USA; 3Department of Medicine, University of Southern California, Los Angeles, CA 90089, USA; 4Department of Urology, University of Irvine, Orange, CA 92868, USA; ncsabhar@hs.uci.edu (N.S.);; 5Department of Urology, SUNY Upstate Medical University, Syracuse, NY 13210, USA; goldberh@upstate.edu

**Keywords:** TURBT, muscle-invasive bladder cancer, bladder preservation, Trimodality treatment, bladder-sparing

## Abstract

**Simple Summary:**

The mainstay treatment of non-metastatic muscle-invasive bladder cancer is radical cystectomy, but in patients who prefer to save their bladder and patients who are at high risk for surgery, bladder-preserving therapies, including tumor removal through complete transurethral resection and chemoradiation (TMT), provide an alternative. However, complete TURBT has significant risks of bleeding, infection, bladder perforation, and tumor cell dissemination, and its necessity in all cases remains unclear. Based on available data, the role of complete TURBT in managing MIBC as a part of a bladder-preserving approach is unclear, and individualized treatment plans and further research are needed to optimize patient outcomes.

**Abstract:**

Radical cystectomy with lymph node dissection and urinary diversion is the gold-standard treatment for non-metastatic muscle-invasive bladder cancer (MIBC). However, in patients who refuse cystectomy, or in whom cystectomy carries a high risk, bladder-preserving therapies remain potential options. Bladder preservation therapies can include maximal debulking transurethral resection of bladder tumor (TURBT), concurrent chemoradiation therapy, followed by cystoscopy to assess response. At this time, maximal TURBT is recommended for patients prior to the initiation of chemoradiation therapy or in patients with residual bladder tumors after the completion of chemoradiation therapy. That being said, TURBT carries significant risks such as bladder perforation, bleeding, and infection, ultimately risking delayed systemic treatment. Hence, understanding its role within trimodal therapy is crucial to avoid undue suffering in patients. Herein, we review the current literature on the impact of debulking TURBT in non-metastatic MIBC.

## 1. Introduction

Muscle-invasive disease is noted in 25% of newly diagnosed bladder cancers. Moreover, up to 50% of patients with high-risk, non-muscle invasive disease may eventually develop invasive disease [1,2,3]. The traditional gold standard treatment for non-metastatic muscle-invasive bladder cancer (MIBC) is cisplatin-based neoadjuvant chemotherapy followed by radical cystectomy (RC), pelvic lymph node dissection, and urinary diversion [4,5]. However, it is known that RC carries significant morbidity and mortality [6,7]. Additionally, studies suggest that urinary diversion may also adversely impact the quality of life in patients [8]. Consequently, bladder-preserving treatment modalities have emerged as favorable options for patients who are either unfit for cystectomy or desire to retain their bladder [9].

The goal of bladder preserving therapy is to minimize the morbidity associated with the cystectomy while also maintaining similarly efficacious oncologic outcomes. Several bladder-sparing options exist, including partial cystectomy, radiation monotherapy, and trimodal therapy (TMT). TMT consists of an initial debulking TURBT, followed by concurrent chemotherapy and radiation. This is followed by regular cystoscopic examination and potentially repeat resection if a recurrent tumor is found [5]. Currently, TURBT is recommended both as a diagnostic tool and a debulking modality. However, TURBT is associated with significant risks, including perforation, tumor cell dissemination, bleeding, infection, and ureteral obstruction; all of which are potential pitfalls if maximal debulking is attempted [10]. An understanding of the importance of TURBT in trimodal therapy is crucial, as minimizing TURBT can potentially mitigate these morbidities and help prevent definitive treatment delay. This review summarizes the current literature on the impact of maximal TURBT in both initial and interval resection following chemoradiation.

## 2. Methods

This comprehensive review conducted on the role of maximal TURBT in bladder preservation strategies involved an extensive literature review, encompassing studies published before 30 April 2024. The search was meticulously executed using specific keywords related to muscle-invasive bladder, TURBT, and trimodal therapy. A systematic literature search was performed in the following databases: MEDLINE, PubMed, and EMBASE databases. Our study focused exclusively on English-language studies.

Studies not pertinent to the topic, such as case reports, were excluded. This review compiles and presents the studies’ findings to provide a comprehensive overview of the current knowledge on the role of TURBT in the management of non-metastatic MIBC.

## 3. Trimodal Therapy (TMT) for Muscle-Invasive Bladder Cancer

### 3.1. Current Standards and Guidelines

The AUA, EAU, and NCCN guidelines have recommended TMT as an alternative to RC for patients with MIBC who are either unfit for RC or for those who desire bladder-preserving therapy [5]. Regardless of the desired approach, all current guidelines strongly emphasize the importance of maximal debulking of the tumor by TURBT.

TMT consists of complete tumor debulking and radiotherapy +/− chemotherapy with the goal of achieving maximal local tumor control. Various options including neoadjuvant chemotherapy, followed by TURBT, radiation monotherapy, or partial cystectomy have been previously introduced as possible bladder preservation strategies. However, they all have had inferior oncologic outcomes when compared with RC or TMT [10,11,12,13]. That being said, in cases where strict TMT is not feasible, the combination of TURBT plus chemotherapy has demonstrated superior cancer-specific survival outcomes over TURBT plus radiotherapy [14].

Appropriate patient selection is the cornerstone for the success of the TMT option. Factors such as carcinoma in situ (CIS), tumor histology, hydronephrosis, bladder function, and the likelihood of successful complete tumor resection should each be considered in order to optimize the patient’s quality of life and oncologic outcomes [15]. One critical group is patients with NMIBC who are Bacillus Calmette–Guerin (BCG)-unresponsive. Despite receiving optimal intravesical immunotherapy, up to 40% of patients experience disease progression or recurrence within two years. A retrospective study by Ferro et al. highlights several predictive factors for BCG failure, including tumor multifocality, lymphovascular invasion, and high-grade disease on restaging TURBT [16]. Patients with pure CIS face a high risk of disease progression even after BCG therapy. Elderly patients, especially those over 70, are at a particularly elevated risk due to diminished immune response to BCG [17]. There are mixed results from prior studies regarding the differences in survival and rate of salvage cystectomies among patients with urothelial carcinomas when compared to variant histologies. For example, a study by Barletta et al. suggests that patients with squamous cell carcinoma may have worse survival outcomes when compared to urothelial carcinoma [18].

Patients should be advised about the potential risks and benefits of TMT in a multidisciplinary setting involving urologists, medical oncologists, and radiation oncologists. It is imperative that patients also fully understand the unique requirement for life-long bladder monitoring, in order to ensure their active participation in the decision-making process [19].

### 3.2. Chemotherapy Regimens in TMT

Two types of TMT have been described before: the split course and the continuous course. The split course involves debulking TURBT followed by concurrent cisplatin-based chemotherapy and radiation. In this case, restaging TURBT mid-treatment would then determine if chemoradiation should be continued or if a radical cystectomy is indicated. In a continuous course, chemoradiation will be delivered after maximal TURBT. This is followed by a restaging TURBT 1–3 months post-chemoradiation in order to assess the tumor response and determine if there is a need for cystectomy [20].

Multiple concurrent chemotherapy options have been shown to be effective, though evidence most strongly supports the use of either cisplatin or mitomycin-C/5-FU-based regimens [21]. Alternative chemotherapeutic options for cisplatin-ineligible patients include single-agent gemcitabine, capecitabine, and hypoxia modification with carbogen/nicotinamide [19,22].

Both twice-a-day radiation with 5-FU/cisplatin and once-daily radiation with gemcitabine have demonstrated greater than 75% 3-year distant metastasis-free survival [23]. Hoskin et al. also showed increased overall survival with radiation and concurrent carbogen/nicotinamide compared to radiation alone [48% vs. 35%] in patients with muscle-invasive bladder cancer with incomplete resection [22].

### 3.3. Comparative Outcomes of RC and TMT

There are no RCTs that directly compare RC and TMT in patients with MIBC. Nonetheless, retrospective, systematic reviews and meta-analyses have consistently demonstrated similar outcomes in selected patients. A systematic review of 57 studies [*n* = 30,293] reported a 10-year overall survival (OS) of 30.9% and 35.1% for TMT and RC, respectively. These studies also revealed a 10-year disease-specific survival (DSS) of 43.5% and 43.1% for higher-stage disease (T3-T4) in patients undergoing TMT or RC, respectively. Previous studies have also demonstrated that 5-year cancer-specific survival (CSS) and OS ranged from 50–84% and 36–74%, respectively [21,24,25,26,27]. A study by Giacalone et al. showed that patients who underwent TURBT followed by concurrent chemoradiation had an improvement in complete response (CR) and 5-yr CSS over different eras [2005–2013 vs. 1986–1995]. They also showed a correlation of visibly complete TURBT with better outcomes [26].

A sub-analysis of Radiation Therapy Oncology Group (RTOG) trials revealed similar outcomes between complete and near-complete responders after the induction phase of bladder-preserving combination therapy for MIBC [28]. Additionally, a recent retrospective multi-institutional analysis of 722 patients by Zlotta et al. demonstrated similar outcomes in metastasis-free survival (MFS), CSS, disease-free survival (DFS), and OS when comparing RC and TMT [29].

It has been shown that most recurrence after TMT is non-invasive and can be addressed conservatively [21]. Salvage cystectomy rates after TMT were reported to be 10–30% [21,24,25,26,27]. Late complications were found to be slightly more frequent, though overall acceptable, in patients who underwent salvage cystectomy when compared to primary RC [30].

Quality of life post-TMT is often better than post-cystectomy, which is an appealing and promising aspect of this treatment option. However, prospective validating studies are needed to establish which patients are likely to receive the most benefit [31]. An essential point in TMT is the need for lifelong bladder monitoring due to the potential of recurrence, even after a complete response.

## 4. Role of Maximal TURBT in Trimodality Treatment

Maximal TURBT is a recommendation per the AUA guideline [5]. In a pooled analysis of six RTOG bladder-preservation studies, Mak et al. found that complete debulking with removal of all visualized tumors was associated with a higher CR rate on multivariate analysis [32]. Likewise, Efstathiou et al. found that resection of all visible tumors was associated with a 79% CR rate vs. 57% in those with incomplete resection of visible tumors [33]. Moreover, complete TURBT has also been found to be a predictor for bladder-intact disease-specific survival [26]. A study by Erlangen et al. also found that complete resection of MIBC after primary TURBT, along with early tumor stage, was the most important predictor of complete response and survival in TMT [34]. A study by Pak et al. compared complete vs. incomplete TURBT prior to neoadjuvant chemotherapy in a cohort of 93 patients with MIBC. They found that complete TURBT prior to NAC was associated with superior overall survival [35].

Evidence supporting maximal resection comes from data suggesting a benefit of a second TURBT. In a study of 90 patients, Suer et al. found that a second TURBT prior to initiating TMT was associated with a statistically significantly higher 5-year DSS [68% vs. 41%] as well as increased OS (63.7% vs. 40.1%), although this did not reach statistical significance [36].

A study by Mak et al. [32] revealed that the presence of hydronephrosis, higher clinical stage T (3/T4), and visibly incomplete TURBT are predictors of worse disease-specific survival on univariate analysis. Only visibly complete TURBT remained a significant predictor of disease-specific survival on multivariable analysis (HR 0.49 (0.25–0.96), *p* = 0.04)). However, it is unclear whether a visually complete TURBT before TMT is truly a prognostic factor or, rather, a surrogate marker of a lower local tumor stage.

James et al. [23] conducted a multicenter, phase 3 trial (BC2001) of 360 patients, comparing the impact of synchronous chemotherapy (mitomycin C and fluorouracil) with radiotherapy vs. radiation therapy alone [23]. They found superior locoregional DFS at two years (67% vs. 54%) and a superior 5-year overall survival rate (48% vs. 35%) in the chemoradiation group when compared to radiation monotherapy. In a long-term follow-up [with a median follow-up of 9.9 years], the rate of salvage cystectomy was reported to be 14% [37]. It is worth noting that most of the patients in this study did not undergo maximal debulking. Yet, despite this, the study demonstrated good oncologic outcomes.

## 5. Radical TURBT as Monotherapy for Bladder Preservation

Radical TURBT involves maximal tumor debulking with the aim of complete resection. Maximal TURBT alone has been associated with up to a 15% chance of pT0 disease at the time of cystectomy [38]. In 2001, it was shown that the 10-year survival probability of patients with MIBC in initial TURBT and T0 or T1 disease on repeat TURBT was the same with either surveillance or radical cystectomy. In that study, 24% of patients treated with TURBT alone died of their disease.

In a separate study looking at outcomes in patients with MIBC, 34 of 99 patients treated with only radical TURBT relapsed in the bladder, with 53% of patients needing a RC [39]. The outcomes of TURBT monotherapy have also been evaluated in a small retrospective cohort of MIBC patients with no evidence of residual disease on re-TURBT. It was reported that 70% of patients who pursued bladder-sparing TURBT were able to preserve their bladder. However, delayed cystectomy was shown to be associated with more advanced disease [positive lymph node disease, pT3 b] [40].

Solsona et al. followed 133 patients with MIBC [most of the tumors were solitary (85%) and without carcinoma in situ (76%) for 15 years after radical TURBT monotherapy [41]. They showed a progression-free survival rate of 75.5%, 64.9%, and 57.8% at 5, 10, and 15 years respectively. 30% of patients progressed, and 7.7% ended up with metastatic disease.

Presently, EAU strongly recommends against offering monotherapy with TURBT as a curative option for MIBC as most patients will not benefit from it [15].

## 6. The Role of Repeat TURBT in Bladder Preservation

A study by Zamboni et al. evaluated 433 patients with complete TURBT prior to RC and showed that 53% of patients had muscle-invasive disease at the time of RC, which is particularly noteworthy. However, the study failed to establish a clear correlation between oncologic outcomes and complete resection, despite the association of pT3-T4 disease by incomplete TURBT [42].

Among the cT0 MIBC patients following re-TURBT (50% of which had neoadjuvant chemotherapy), only 35.7% remained pT0 at the time of RC [43]. They reported 24.8% with at least pT3 or nodal disease, leading to significantly worse RFS and OS. These findings were further supported by another study on MIBC patients who underwent TURBT after NAC and before RC, revealing a 32% rate of false downstaging [44].

Previous studies have underscored the importance of the absence of disease on re-TURBT specimens as a predictor of improved prognosis. However, it has been shown that the resection of all visible tumors prior to neoadjuvant chemotherapy is not associated with pT0 disease at the time of cystectomy [45]. The study of 433 pT0 cases at the time of RC by Chromecki et al. failed to establish the role of maximal TURBT prior to NAC (*p* = 0.13). An analysis of the adjusted relative risk estimated that 38% of the pathological response seen at the time of RC among patients receiving NAC can be attributed to the TURBT alone [46].

Recent studies have also failed to establish an association between complete TURBT and survival outcomes after RC. Ghandour et al. [47] conducted a retrospective review of 100 patients who received NAC followed by RC for MIBC (49 patients with complete TURBT and 51 patients with incomplete resection) and found that grossly complete TURBT is not particularly associated with ypT0 or survival outcomes.

Another recent study of 548 patients with MIBC (using propensity scores to match patients treated with and without re-TURBT before RC) showed that 37.5% of cT2 patients had extravesical disease at RC. The absence of disease on re-TURBT specimens was associated with improved prognosis, but no difference in survival outcomes was observed [48].

Importantly, in a cohort of 153 patients, of which 76% had complete TURBT, investigators discovered no significant difference in achieving ypT2N0 or ypT0N0 based on the completion of TURBT. Furthermore, they observed no significant association of complete tumor resection with oncologic outcomes after a median follow-up of about four years. However, the hazard of death from any cause, even after adjustment for ECOG and pT stage, was significantly higher in the incomplete resection group [49]. These findings strongly support the conclusion that re-TURBT is not entirely necessary to achieve a visibly complete resection before NAC and RC.

A study led by Alsyouf et al. examined 115 NMIBC patients who underwent blue light TURBT followed by a restaging TURBT [50]. Contrary to expectations, they found that using blue light did not significantly reduce the rate of residual disease. This result underscores the limitations of even new modalities like blue light, which are touted for their ability to provide more precise resection. It is clear that reaching pT0, even for low-stage disease, remains a challenge.

Table 1 summarizes key studies evaluating the impact of maximal TURBT on outcomes in patients with muscle-invasive bladder cancer.

## 7. Key Challenges and Considerations in Complete TURBT: Risks and Complications

### 7.1. Quality of TURBT Resection Specimens

TURBT has been the preferred resection technique for decades given its relatively low morbidity. However, it is now known that this procedure is not without risks. Potential complications include tumor cell dissemination, hematuria, bladder spasms, urinary retention, urinary tract infection, and bladder perforation. Complication rates reported are approximately 4–6%, with urinary tract infections and significant hematuria as the most common. Bladder perforation, which requires surgical repair and has a risk for extravesical tumor seeding, occurs in 0.5 to 8.3% of cases [51,52].

The presence of detrusor muscle is considered a surrogate for resection quality, except in low-grade or non-invasive tumors. Chamie et al. found that detrusor muscle was present in only 52% of TURBT cases, regardless of stage or grade, and its absence was associated with increased five-year mortality [53]. Previous studies found that surgeon experience was independently associated with detrusor muscle presence in TURBT specimens, and both the absence of detrusor muscle and resection by junior surgeons were linked to higher recurrence rates at first follow-up cystoscopy [54,55,56,57]. Another study confirmed that TURBTs performed by residents were associated with a lower likelihood of finding detrusor muscle in the TURBT specimens [58].

Recently, Volz et al. reviewed 2058 cases and found detrusor muscle was found in approximately one out of three cases [59]. Longer surgery duration was an independent predictor for a lower likelihood of detrusor muscle presence. Other significant risk factors for missing detrusor muscle included papillary tumors and tumor localization in the bladder dome or posterior bladder wall. The absence of detrusor muscle in high-grade bladder cancer is correlated with reduced recurrence-free survival (RFS).

Extensive bladder tumors can lead to inevitable incomplete resection which results in prolonged procedures with increased risks and complications [60]. Adequate staging by TURBT can be challenging and may lead to consequences such as bladder perforation, delayed treatment, and the risk of tumor cell dissemination. Additionally, the financial and healthcare system burdens cannot be understated [61].

### 7.2. Risk of Tumor Cell Dissemination

Theoretically, TUBRT could lead to the dissemination of tumor cells into circulation. The poorer inter-cellular adhesion in higher-grade tumors and deeper resection for invasive tumors have been thought to be risk factors predisposing these patients to higher systemic spread of cancer cells during TURBT. That being said, prior studies have not found tumor cell dissemination into peripheral circulation following TURBT. Antoniewicz et al. evaluated the expression of 14 gene types by quantitative RT-PCR on RNA isolated from peripheral blood samples at 1, 3, 7, and 30 days after surgery [62]. Similarly, a study by Desgrandchamps et al. also failed to demonstrate the detection of tumor cells following TURBT [63]. The superficial nature of most tumors and small samples in this study could be the limitation of demonstrating circulating tumor cells [CTCs].

The collection of blood samples from the IVC both before and during TURBT after placement of an IVC catheter showed an increase in the CTCs intraoperatively [64]. Blaschke et al. demonstrated a rise in CTCs following TURBT in patients with high-grade and muscle-invasive disease [65]. A meta-analysis by Zhang et al. revealed that the presence of CTCs in patients with urothelial cancer, irrespective of the stage at the time of diagnosis, is an independent predictor of poor outcomes [66]. Furthermore, it has been shown that even a single CTC can predict shorter cancer-specific and overall survival in non-MIBC patients [67].

More recently, tumor cell dissemination during TURBT has been demonstrated in two studies. Nayyar et al. found a measurable rise in CTCs post-TURBT in 16.98% of patients, with 52.94% of those having had muscle-invasive disease [68]. Despite this finding, it was not specifically associated with adverse oncological outcomes. Similarly, Haga et al. observed a significant increase in postoperative CTC count in the MIBC group than in the non-MIBC group, suggesting that deeper resection and excessive infusion pressure should be avoided in MIBC patients [69].

## 8. Future Directions beyond Complete TURBT: Advancing Bladder Cancer Treatment through Emerging Therapies and Innovations

### 8.1. Genomic Approaches in Bladder Preservation: Personalized Treatment

The movement towards advanced molecular techniques, including detecting specific genomic alterations in tumor tissue, has shaped the advancement of oncology. The RETAIN Trial is a single-arm, phase 2, non-inferiority trial that utilizes information gathered from genomic alterations and clinical response to TURBT followed by chemotherapy for a risk-adapted approach to bladder sparing. This study selected patients with MIBC who had mutations in the genes, ATM, ERCC2, FANCC, or RB1 and achieved a complete clinical response to dose-dense MVAC chemotherapy. These patients underwent active surveillance and were followed for evidence of progression. The primary endpoint was metastasis-free survival at two years. With 71 patients enrolled, the 2-year metastasis-free survival was 72%, which did not meet the prespecified non-inferiority condition [70]. Nonetheless, with a median follow-up of 41 months, 50% of patients were able to avoid cystectomy without metastatic disease. This suggests that further studies are necessary to refine the target patients appropriate for this treatment approach.

The Alliance trial is an ongoing study actively recruiting with a goal of 271 patients with T2-T4aN0/xM0 MIBC [71]. Those with DDR gene mutations who achieve a clinical response of less than T1 after neoadjuvant cisplatin-based chemotherapy are eligible for surveillance. Patients who do not achieve a response of less than cT1 or do not have DDR mutations will undergo either RC or TMT. The results of this trial will hopefully provide further information about the role of decision-making in treatment plans based on genomic studies.

### 8.2. Transforming the Future of Bladder Cancer Treatment with Immunotherapy: A New Era of Precision Care

In the future, we may see the incorporation of immunotherapy into the existing chemotherapy regimens. This promising development could potentially lead to improved oncologic outcomes.

De Ruiters et al., in a phase 1 study, investigated the addition of concurrent immune checkpoint inhibition to chemoradiation for bladder preservation and found high rates of metastasis-free and overall survival [72].

Notably, the INTACT trial, a phase 3 study that explores the role of atezolizumab in conjunction with chemoradiation, has recently concluded its enrollment phase [73]. This trial is expected to provide crucial data that will further our understanding of this field.

### 8.3. MRI in Bladder Preservation: Emerging Data and Impact

mpMRI of the bladder has the ability to detect muscle-invasive disease and has the potential to eliminate the need for staging TURBT. A recent consensus, developed by a panel of experienced professionals, including radiologists, urologists, oncologists, radiation oncologists, and patient advocacy representatives, has underscored the pivotal role of MRI in bladder cancer [74]. The panel recommended interpreting MR images according to the VI-RADS guideline and always performing MRI before TURBT, if available.

The use of mpMRI with VI-RADS is recommended to differentiate T1 and T2 disease in bladder cancer [4]. The potential of a modified version of VI-RADS, called nacVI-RADS, to predict treatment response and pre-operative outcomes is a promising development in the treatment of bladder cancer [75]. Radiomic-based techniques in the future could predict muscle invasion and eliminate the need for radical TURBT. For example, a VI-RADS score of ≥3 indicates MIBC, with high sensitivity, specificity, and AUC, offering hope for more accurate diagnoses [76,77,78,79,80].

VI-RADS is integral as it helps to identify patients likely to benefit from neoadjuvant chemotherapy and bladder-sparing therapy with chemoradiation. For those who are deemed less likely to benefit from bladder-sparing methods, VI-RADS can also aid in planning for radical, complete TURBT [81,82,83]. Six recent studies reviewed by Klempfner et al. indicate that VI-RADS scoring accurately predicts muscle invasion and successfully aids in NMIBC/MIBC differentiation [84].

mpMRI preceding TURBT may lead to earlier MIBC recognition and treatment initiation [85,86,87,88]. The PURE-01 trial evaluated mpMRI for predicting clinical response to neoadjuvant pembrolizumab, with internal validation detecting pT0 in 62% of patients and external validation in 73% of patients [89].

Taguchi et al. proposed an innovative bladder cancer management algorithm that leverages the high sensitivity of VI-RADS ≥ 3 and the high specificity of VI-RADS ≥ 4 to avoid unnecessary procedures and expedite care [61]. An example of this is that patients with VI-RADS scores of 2 or less in this algorithm could potentially avoid re-TURBT, particularly when initial specimens lack detrusor muscle. Alternatively, patients with a VI-RADS score of 4 or higher might proceed directly to radical cystectomy, circumventing the need for deep resection or re-TURBT.

Preliminary data from the BladderPath study suggests that mpMRI accurately stages bladder cancer and may help avoid TURBT in selected patients [90]. The study screened 279 patients and randomized 113, with initial experiences indicating that mpMRI could feasibly direct possible patients with MIBC for staging instead of TURBT. A 5-point Likert scale accurately identifies patients with low risk of MIBC, and flexible cystoscopy biopsies appear sufficient for diagnosing bladder cancer.

In Table 2, we highlight the emerging approaches and future directions in the management of MIBC, including genomic strategies, immunotherapy, and advanced imaging techniques, along with key findings from recent trials and studies.

## 9. Proposed Shift in TURBT: From Complete Resection to Diagnostic Biopsy for Bladder Preservation

Bladder preservation is a viable alternative for very selective patients with MIBC who are either unsuitable for RC or prefer bladder-sparing approaches. Based on the data reviewed in this manuscript, we propose an algorithm that optimizes oncological outcomes while minimizing treatment-related morbidity. The initial step involves thorough patient evaluation, including clinical staging via cystoscopy and imaging (preferably multiparametric MRI using the VI-RADS scoring system) to assess the extent of the disease. Tumor characteristics such as histological subtype, presence of CIS, and patient-specific factors, including comorbidities, renal function, and performance status, should guide decisions about bladder preservation versus radical cystectomy. Genomic profiling, when available, can provide crucial insights into patient suitability for bladder-sparing therapies, especially when mutations in DNA damage response DDR genes like ATM or ERCC2 are detected. We propose a shift in the traditional approach to TURBT, which is currently suggested by guidelines. Rather than emphasizing maximal TURBT, which is associated with significant morbidity such as bleeding, perforation, and tumor cell dissemination, TURBT should be employed primarily for biopsy and diagnostic purposes. The focus should be on obtaining adequate tissue for staging and histopathological evaluation rather than attempting complete resection of the tumor. This more conservative approach could minimize patient suffering while preserving critical oncological outcomes, particularly when integrated into TMT, which combines limited TURBT with chemoradiation.

Following diagnostic TURBT, patients can proceed directly to TMT, which offers comparable survival outcomes to RC in selected patients. By reducing the emphasis on maximal tumor resection, this strategy lowers the risks of surgical complications and streamlines the treatment process. Future research should focus on refining patient selection criteria and treatment protocols to further optimize the role of TURBT, restricting it primarily to biopsy purposes while enhancing the efficacy of TMT as a bladder preservation strategy.

## 10. Conclusions

In conclusion, while maximal TURBT plays a significant role in the initial management and interval assessment in TMT for MIBC, its necessity in all cases is still under debate. The choice of treatment should be individualized based on patient-specific factors and the presence of adverse prognostic indicators. Further studies are needed to establish standardized protocols, refine patient selection, and optimize patient outcomes.

## Figures and Tables

**Table 1 cancers-16-03361-t001:** Summary of Studies Evaluating the Impact of Maximal TURBT on Outcomes in Muscle-Invasive Bladder Cancer (MIBC).

Study	Patient Population	TURBT Approach	Outcomes	Key Findings
Efstathiou et al., 2012 [33]	348 patients with MIBC	Complete TURBT	CR: 79% for complete resection vs. 57% for incomplete resection	Complete TURBT associated with better CR and survival rates
James et al., 2012 [21]	360 patients with MIBC	Majority did not undergo maximal debulking	Superior 5-year OS (48% vs. 35%) with chemoradiation vs. radiation alone	Good outcomes despite lack of maximal TURBT in most patients
Mak et al., 2014 [32]	468 patients with MIBC (T2-T4)	Complete vs. Incomplete TURBT	the presence of hydronephrosis, higher clinical stage (T3/T4), and visibly incomplete TURBT are predictors of worse disease-specific survival on univariate analysis.	Only visibly complete TURBT remained a significant predictor of disease-specific survival on multivariable analysis. It is unclear whether a visually complete TURBT before TMT is truly a prognostic factor or a surrogate marker of a lower local tumor stage.
Suer et al., 2016 [36]	90 patients with MIBC	Second TURBT	Higher 5-year DSS (68% vs. 41%) and OS (63.7% vs. 40.1%)	Second TURBT significantly improves 5-year DSS and should be performed in patients with MIBC who are going to be treated with bladder-preserving protocols.
Zamboni et al., 2019 [42]	433 patients with MIBC undergoing RC	Complete TURBT	53% had MIBC at RC; no clear correlation with oncologic outcomes	Complete TURBT not clearly associated with improved outcomes.
Pak et al., 2021 [35]	93 patients with MIBC before neoadjuvant chemotherapy (NAC)	Complete vs. Incomplete TURBT	Superior survival for complete TURBT prior to NAC	Complete TURBT before NAC linked to better survival outcomes

**Table 2 cancers-16-03361-t002:** Emerging Approaches and Future Directions in the Management of Muscle-Invasive Bladder Cancer.

Approach	Study/Trial	Patient Population	Key Findings	Clinical Implications
Genomic Approaches	RETAIN Trial [70]	71 patients with MIBC and ATM, ERCC2, FANCC, or RB1 gene mutations that achieved a complete clinical response to dose-dense MVAC chemotherapy after TURBT	Achieved a 2-year metastasis-free survival rate of 72%, but did not meet the prespecified non-inferiority condition. 50% of patients avoided cystectomy without developing metastatic disease.	Demonstrates the potential for risk-adapted bladder preservation strategies using genomic profiling.
Genomic Approaches	Alliance A031701 Trial [71]	Target: 271 patients with T2-T4aN0/xM0 MIBC	Ongoing trial evaluating the role of DDR gene mutations in guiding bladder preservation versus cystectomy decisions.	Aims to establish whether a genomic-guided approach can safely reduce the need for cystectomy in selected patients.
Immunotherapy in Bladder Preservation	De Ruiters et al., 2022 [72]	Phase 1 study with MIBC patients	High rates of metastasis-free survival and overall survival when nivolumab +/− ipilimumab was added to chemoradiotherapy.	Suggests that immune checkpoint inhibitors may enhance the efficacy of bladder preservation protocols.
Immunotherapy in Bladder Preservation	INTACT (S/N1806) Trial [73]	73 patients in the safety update	Ongoing Phase 3 trial; preliminary safety data indicates tolerability of adding atezolizumab to chemoradiotherapy.	The trial could establish a new standard of care by integrating immunotherapy into bladder-preserving treatments.
MRI in Bladder Preservation	BladderPath Study [90]	279 patients screened, 113 randomized	Early results suggest mpMRI can accurately stage bladder cancer, potentially reducing the need for TURBT in initial evaluation.	Could shift the standard of care towards imaging-based staging, minimizing invasive procedures in selected patients.
MRI in Bladder Preservation	PURE-01 Trial [89]	82 Patients with MIBC undergoing neoadjuvant pembrolizumab	In post-pembrolizumab muscle-invasive bladder cancer, mpMRI sequence assessment had acceptable interobserver variability.	Supports the use of mpMRI as a non-invasive tool for monitoring response to neoadjuvant immunotherapy.

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
