# Peer review of "The Role of Maximal TURBT in Muscle-Invasive Bladder Cancer: Balancing Benefits in Bladder Preservation and Beyond"

_cancers, 2024, doi:10.3390/cancers16193361_

Round 1

Reviewer 1 Report

Comments and Suggestions for Authors

The authors provide a very nice review of the value of TURBT in the treatment pathway of muscle invasive bladder cancer.

The title is misleading as only a portion of the manuscript is focused on the TMT and the remainder of the manuscript does not directly apply the data provided to TMT.  With this, the authors could create a title for the review manuscript that is more suitable for the content provided or rewrite the manuscript with more focus on TMT.

The authors have done a good job on summarizing key references on the topic of TURBT in muscle invasive bladder cancer.  It would be nice to see them take the next step and propose the best algorithm to be used based on the data provided.  Is this in alignment with current guidelines?  Should the current guidelines be changed?  What additional data or trials are needed?

Author Response

We sincerely thank you for your thoughtful review of our manuscript. Your comments were insightful and have helped us refine our work. Below, we address each of your points and explain the changes made:

Reviewer’s Comment 1: "The title is misleading as only a portion of the manuscript is focused on the TMT and the remainder of the manuscript does not directly apply the data provided to TMT. With this, the authors could create a title for the review manuscript that is more suitable for the content provided or rewrite the manuscript with more focus on TMT."

Response: We appreciate your feedback regarding the title. To better align with the manuscript's content, we have revised the title to: “The Role of Maximal TURBT in Muscle-Invasive Bladder Cancer: Balancing Benefits in Bladder Preservation and Beyond.” This updated title more accurately reflects the review's scope, including the role of TURBT in TMT and its implications in bladder preservation.

Reviewer’s Comment 2: "The authors have done a good job of summarizing key references on the topic of TURBT in muscle-invasive bladder cancer. It would be nice to see them take the next step and propose the best algorithm to be used based on the data provided. Is this in alignment with current guidelines? Should the current guidelines be changed? What additional data or trials are needed?"

Response: In response to your insightful suggestion, we have added a new section titled “Proposed Shift in TURBT: From Complete Resection to Diagnostic Biopsy for Bladder Preservation.” This section outlines a new approach for using TURBT in bladder preservation strategies. Furthermore, we have highlighted specific areas where additional studies are necessary to validate this proposed shift. These additions provide a clearer direction for the future of TURBT in muscle-invasive bladder cancer and align with the review’s goal of offering practical and forward-looking insights.

We trust these revisions have addressed your comments and are grateful for your valuable input.

Best regards

Reviewer 2 Report

Comments and Suggestions for Authors

This review si well written but adds nothing to the state of the art, mostly a collection of retrospectivr studies.The paper doesnt have the strenght to cast doubt on what the current guidelines  suggest regarding    TMT  for MIBC

Author Response

We would like to thank you for your thoughtful review of our manuscript. We have taken your comments into careful consideration and have made revisions to strengthen the manuscript. Below, we address your points:

Reviewer’s Comment: "This review si well written but adds nothing to the state of the art, mostly a collection of retrospectivr studies.The paper doesnt have the strenght to cast doubt on what the current guidelines  suggest regarding    TMT  for MIBC."

Response: We appreciate your recognition of the writing quality and understand your concern about the need for more novel insights. To address this, we have revised the manuscript to more clearly articulate our proposed shift in the role of TURBT in MIBC, specifically in relation to TMT. In response, we have added a new section titled “Proposed Shift in TURBT: From Complete Resection to Diagnostic Biopsy for Bladder Preservation.” In this section, we outline an alternative approach to TURBT in the context of bladder preservation that challenges the current emphasis on maximal resection. We propose that TURBT could primarily serve as a diagnostic tool rather than as a complete therapeutic intervention, particularly in patients undergoing TMT.

This approach offers a fresh perspective on the role of TURBT and highlights areas where further research and updates to guidelines may be warranted. While much of the data we reference comes from retrospective studies, these studies form the basis of our argument, and we have emphasized the need for prospective trials to validate our proposed approach.

We believe this revision strengthens the manuscript’s contribution to the field by proposing a significant shift in clinical practice and suggesting future directions for research.

We hope these revisions address your concerns, and we appreciate your constructive feedback.

Sincerely,

Reviewer 3 Report

Comments and Suggestions for Authors

Bladder cancer (BCa) is the most frequent malignant carcinoma of the genitourinary tract, being the 10th tumor for incidence when considering both sexes, and the 7th considering only the men population. Up to 70–85% of patients with BC are initially diagnosed as non-muscle invasive bladder cancer (NMIBC), while 15–30% are diagnosed as or progress to muscle-invasive bladder cancer (MIBC).

The initial treatment strategy of patients with high-grade NMIBC is the transurethral resection of bladder tumor (TURBT), followed by intravesical therapy of bacillus Calmette–Guerin or chemotherapy (according to grade and focality of the tumor), while, in MIBC, neoadjuvant chemotherapy followed by radical cystectomy (RC) with bilateral pelvic lymphadenectomy represents the mainstay of treatment. Despite this, not every patient can undergo an RC due to clinical contraindications or just because they refuse it. For that reason, bladder-preserving treatment strategies come out as an alternative favorable option for these patients, minimizing the morbidity associated with the cystectomy while also maintaining similarly efficacious oncologic outcomes. That is why the authors focused their attention in this article on a comprehensive review conducted on the role of TURBT in trimodal therapy.

The authors should be congratulated on the interesting topic discussed. Numerous steps have been made in the treatment of this condition over the years, but 

The study has sufficient merit to be considered for publication, although major revisions are required. 

1.     Methods and methodology are robust.

2.     Results and conclusions are well presented.

3.     Tables and graphics are clearly described.

4.     The authors focused a lot on the role of bladder inflammation as a risk factor for tumor onset and progression. I suggest providing more detailed information about new cohorts of patients that can undergo RC, such as BCG-unresponsive and pure carcinoma in situ ones: please, add some information about it. These two papers offer a valid solution for the purpose (https://doi.org/10.1016/j.urolonc.2022.05.016https://doi.org/10.1016/j.clgc.2021.12.005). A lecture is suggested.

Comments on the Quality of English Language

Minor editing.

Author Response

We would like to express our sincere gratitude for your positive and constructive review of our manuscript. We have carefully considered your feedback and made the following revisions:

Reviewer’s Comment 1: "The authors focused a lot on the role of bladder inflammation as a risk factor for tumor onset and progression. I suggest providing more detailed information about new cohorts of patients that can undergo RC, such as BCG-unresponsive and pure carcinoma in situ ones: please, add some information about it. These two papers offer a valid solution for the purpose."

Response: We appreciate your suggestion and have now included detailed information regarding BCG-unresponsive patients and those with pure carcinoma in situ. This addition is found in the section titled "Current Standards and Guidelines," where we have discussed how these patient cohorts fit into the treatment paradigm. We have also incorporated the two references you recommended (DOIs: 10.1016/j.urolonc.2022.05.016, 10.1016/j.clgc.2021.12.005) to provide further insights into the evolving treatment strategies for these challenging cases.

By doing so, we have enriched the manuscript with a more comprehensive perspective on the management of patients who might undergo radical cystectomy while exploring bladder-preserving strategies for those who are BCG-unresponsive or present with carcinoma in situ.

Reviewer’s Comment 2: "Methods and methodology are robust. Results and conclusions are well presented. Tables and graphics are clearly described."

Response: We are pleased that you found the methodology, results, and presentation to be strong, and we have maintained these aspects of the manuscript. We are grateful for your acknowledgment.

We hope these revisions have addressed your concerns. Thank you for your valuable input, which has improved the overall quality of the manuscript.

Sincerely,

Round 2

Reviewer 2 Report

Comments and Suggestions for Authors

ok in present form

Reviewer 3 Report

Comments and Suggestions for Authors

Authors answered all comments and suggestions

Comments on the Quality of English Language

Minor editing